# Smart Bacteria-Responsive Drug Delivery Systems in Medical Implants

**DOI:** 10.3390/jfb13040173

**Published:** 2022-10-01

**Authors:** Yijie Yang, Xue Jiang, Hongchang Lai, Xiaomeng Zhang

**Affiliations:** Shanghai Key Laboratory of Stomatology, Department of Oral and Maxillo-Facial Implantology, Shanghai Ninth People’s Hospital, School of Medicine, Shanghai Jiao Tong University, Shanghai 200011, China

**Keywords:** anti-bacterial, implants, drug release, stimuli-responsive

## Abstract

With the rapid development of implantable biomaterials, the rising risk of bacterial infections has drawn widespread concern. Due to the high recurrence rate of bacterial infections and the issue of antibiotic resistance, the common treatments of peri-implant infections cannot meet the demand. In this context, stimuli-responsive biomaterials have attracted attention because of their great potential to spontaneously modulate the drug releasing rate. Numerous smart bacteria-responsive drug delivery systems (DDSs) have, therefore, been designed to temporally and spatially release antibacterial agents from the implants in an autonomous manner at the infected sites. In this review, we summarized recent advances in bacteria-responsive DDSs used for combating bacterial infections, mainly according to the different trigger modes, including physical stimuli-responsive, virulence-factor-responsive, host-immune-response responsive and their combinations. It is believed that the smart bacteria-responsive DDSs will become the next generation of mainstream antibacterial therapies.

## 1. Introduction

Recently, the rapid development of implantable biomaterials has benefited people suffering from bone and dentition defects. However, all the surgical interventions that involve implantation of biomaterials face the risk of failure due to aseptic loosening and bacterial infections. The high predisposition for infections around post-implant sites is caused by lowered immune system efficacy and the adhesion and biofilm-forming ability of bacteria. Biofilms, in which bacteria are protected from the immune responses, dynamic environments and conventional antibiotics, are essential for the proliferation of bacteria [1]. The common treatments for peri-implant infections are limited to a combination of aggressive surgical debridements and systemic antibiotic regimens, and may eventually end up with device removal if there is no way to control the infections. Moreover, a key feature of bacterial infections is recurrence, which happens in approximately 30% of all cases [2], which indicates that repeated antibiotic treatments are necessary. However, the more frequent antibiotics are used, the higher the probability of antibiotic resistance.

Although some of the biomaterials show antibacterial properties, the majority of antibacterial activity is carried out through drug delivery systems (DDSs). Conventional DDSs load drugs mainly through physically adding large antibiotics to the matrix or covalently attaching them to the surfaces. However, the physically drug-loaded DDSs may provoke the abrupt release of drugs, which is known for its cytotoxicity. The covalently drug-loaded DDSs, in the meantime, limit antibacterial effects to the system surfaces, because of the characteristics of covalent bonds. Furthermore, the common problem with conventional DDSs is that they cannot be administered on demand [3]. The expected pattern of administration within conventional DDSs is generally sustained release of drugs. When infections occur, the local level of antibiotics may fail to reach the effective therapeutic dose according to this delivery pattern, while in the absence of infections, background leakage of antibiotics can exacerbate antibiotic resistance. 

In this context, it is urgent to develop a smart bacteria-responsive DDS that can automatically release antibacterial agents from the implants when infections occur, in a more effective manner without background leakage. Namely, antibiotics should be latent in the absence of bacterial infections, but released adequately to kill bacteria immediately in response to infections. Strategies that utilize the changes specific to the bacteria-infected microenvironment as a unique key to activate drug release have attracted widespread attention in the treatments of peri-implant infections. For instance, bacterial infections can result in an acidic microenvironment (pH = 5.0–5.5) that is distinct from normal physiological conditions (pH = 7.4) [4]. Additionally, the overexpression of virulence factors, such as hyaluronidase, gelatinase and phospholipase, also makes the infected area different from the others [5,6]. Taking those features of infection sites as a stimulus for antibiotic release, a “smart” stimuli-responsive DDS can be designed to achieve more localized and controlled drug release. The greatest benefits of such smart systems are the enhanced efficacy due to higher local concentrations, minimized systemic side effects, and the ability of the released agents to diffuse into the peri-implant tissues, thereby killing bacteria both on the implant surfaces and within the surrounding environment [7,8,9].

Stimuli-responsive materials have been investigated in the biomedical field for several decades, including as DDSs. Here, we summarized a few smart bacteria-responsive DDSs mainly designed to prevent or solve peri-implant infections (Figure 1). The aim of this review is to summarize and analyze the design principles, autonomous reactiveness against bacteria and antibacterial effects of these systems.

## 2. Materials and Methods

### 2.1. Search Strategy and Study Selection Processes

A comprehensive search was conducted via the following medical databases: PubMed, Embase, and Web of Science, for articles published from 1 January 2012 to 18 June 2022 in English. Search terms included (antibacterial OR anti bacterial OR anti-bacterial OR antibacteria OR anti bacteria OR anti-bacteria OR antimicrobial OR anti-microbial OR antibiotics OR antibiotic OR Bacteriocidal OR Bacteriocide OR bacteriocides OR Anti-Mycobacterial OR anti mycobacterial OR Antimycobacterial OR infection OR anti-infection OR infectious) AND (implants OR implant OR prosthesis OR “Prosthetic Implants” OR “Implant, Prosthetic” OR “Implants, Prosthetic” OR “Prosthetic Implant” OR “Implants, Artificial” OR “artificial implants” OR “artificial implant” OR “Implant, Artificial” OR Prostheses OR Endoprosthesis OR endoprostheses OR nanoparticles OR nanoparticle OR nano-particles OR nano-particle) AND (“Delivery System, Drug” OR “Delivery Systems, Drug” OR “Drug Delivery System” OR “System, Drug Delivery” OR “Systems, Drug Delivery” OR “Drug Targeting” OR “Drug Targetings” OR “Targeting, Drug” OR “Targetings, Drug” OR “Drug delivering” OR “Drug release”). The electronic search showed a total number of 16,443 titles, after removing 3354 duplicates. Furthermore, relevant references were manually searched via the reference lists of the included studies and 4 studies were added to the full-text evaluation. A total of 16,294 studies with clearly irrelevant topics and abstracts or ineligible article types were excluded. The final inclusion was based on the inclusion criteria. The study flow diagram is shown in Figure 2. 

### 2.2. Inclusion and Exclusion Criteria

The inclusion criteria for the study selection were as follows: 1Primary studies regarding autonomous bacteria-responsive DDSs.2Studies aiming to eliminate bacteria via releasing antibacterial drugs. 3Studies reporting the detailed data of anti-bacterial assays in vitro or in vivo, such as bacterial inhibition rate (BIR), zones of bacterial inhibition (ZOI) and morphological characterization of bacteria (MCB).

The exclusion criteria for the study selection were as follows: 1Studies that performed controlled drug release by additional artificial activation. 2The DDSs were not designed for antibacterial purposes.3Studies missing detailed data of anti-bacterial assays.

Two researchers (Y.J.Y and X.J) independently conducted the search and screened the titles, abstracts and full text of the papers. Discrepancies were resolved via discussions amongst the researchers. An overview of the experimental details is given in Table 1.

## 3. Results

### 3.1. Physical Stimuli-Responsive Systems

There will be a few physical changes within the infected microenvironment, such as reduced pH and locally elevated temperature. These physical stimuli have already been used to activate the release of antibiotics [56,85,86].

#### 3.1.1. pH-Responsive Systems

The most commonly used trigger is the abnormal change in local pH. As bacterial metabolism produces lactic acid and acetic acid, the local pH, dropping from 7.4 to 6.0 or lower, can be used to trigger the release of antibiotics [4]. Chemical bonds, such as the Schiff base, acetal linkage, and metal ion coordination bonds that are stable under neutral conditions but broken at lower pH, are often utilized to realize the pH-responsive release (Figure 3) [22,31,41].

In recent years, there have been studies on various kinds of pH-responsive polyelectrolyte multilayer films (PEMs), such as poly(acrylic acid) (PAA) [24] and poly(methacrylic acid) (PMAA) [21] films as antibacterial coatings. When a sudden decrease in pH disrupts the original electrostatic equilibrium between weak acidic/alkaline polyelectrolytes and incorporated antibiotics, those PEMs undergo swelling to re-balance the charge, which accomplishes the autonomous release of antibiotics. Chen et al. reported the fabrication of a smart system based on the switchable ability of PMAA as a gating element of pH-stimulated delivery of antimicrobial peptides (AMPs) [21]. The system was able to remain stable and extend the passive release of AMPs to 10 days under physiological conditions, while as the pH decreased, the PMAA collapsed to open the nanotubes and released adequate AMPs to kill the bacteria.

Besides electrostatic attraction, the Schiff reaction, which involves a dynamic covalent imine bond formation via the crosslinking of amine groups and aldehyde groups, is also a promising strategy for smart drug delivery [87]. The Schiff base is pH-responsive according to its chemical structure [88]. Researchers have already proved that the DDS, in which alginate dialdehyde (ADA) was conjugated with gentamicin (GEN) via the Schiff reaction, exhibited superior pH responsiveness and could prevent localized infections both in the early stages (6 h) and in the long term (72 h) [22].

Based on the pH response of metal ion coordination polymers (CPs) on TNTs, a novel smart DDS was designed by Wang et al., triggered by the change in the environment acidity due to *S. aureus* and *E. coli* infections [41]. TNTs were functionalized via amination by 3-aminopropyltriethoxysilane (APTES), into which drugs such as ibuprofen, vancomycin, or silver nitrate were harnessed. The researchers found that the CPs formed by 1,4-bis (imidazole-1-ylmethyl) benzene and Zn2+ or Ag+ could successfully block the drug release from TNTs in a neutral environment and could be triggered to open and release antibiotics once the environment became acidic. The release rate gradually increased as the pH value further decreased, indicating that the DDS was a controllable smart DDS for peri-implant infections.

#### 3.1.2. Temperature-Responsive Systems

As is widely known, bacterial infections will raise the local temperature of the first place, which is also regarded as a trigger. Recently, various smart polymers that undergo a phase transition within a specific temperature range in response to an abrupt change have shown great promise in the aspect of drug delivery. The polymers are characterized by a critical solution temperature (CST), a narrow temperature range in which the hydrophobic/hydrophilic interactions between a polymer chain and aqueous medium change. These changes can lead to either chain collapse or swelling. The polymer with lower critical solution temperature (LCST) shows the solution phase below CST and becomes insoluble or forms hydrogels over CST. The polymers with LCST are mostly used for developing DDSs [89]. For example, poly(N-isopropylacrylamide) (PNIPAM) is one of the most representative smart polymers, transitioning from a two-phase to a one-phase mixture in an aqueous environment as the temperature decreases below a value of 37 °C [90].

Choi et al. reported a temperature-responsive (poly(di(ethylene glycol) methyl ether methacrylate)) (PDEGMA) brush coating that allowed the controlled release of levofloxacin [55]. The localized temperature rising of the infected site triggered the release due to the LCST behavior of the brushes. The antibacterial activity of levofloxacin, as well as the antifouling effects of PDEGMA, suppressed bacterial colonization and biofilm growth, as demonstrated in vivo tests with rats infected with *S. aureus*.

#### 3.1.3. Contact-Responsive Systems

Another alternative involves contact killing. Polycations attached to complex biomaterial surfaces with negatively charged bacterial shells allow the polycations to penetrate the shell and kill the bacteria. Contact killing does not require time for the metabolic processes to achieve threshold levels based on other triggers, such as pH switches. However, the effective range of contact killing is more restricted, in contrast to release mechanisms that rely on diffusion [91].

Recently, a new concept called contact transfer has been introduced [57], which integrates ideas from contact killing and stimuli-responsiveness. It involves the transfer of untethered cationic antibiotics from surfaces of biomaterials to bacteria, when bacteria come close to biomaterials. Liang et al. designed anionic microgels loaded with small-molecule cationic antibiotics based on this novel concept [57]. The release of antibiotics was triggered specifically by bacterial contact, not the contact performed by macrophages or osteoblasts. Thus, the antibacterial property and biocompatibility were ensured. Researchers concluded that the negative charge and hydrophobicity of the bacterial envelope changed the local thermodynamic equilibrium that controlled antibiotic–microgel complexation, leading to antibiotics release.

Most of the DDSs mentioned above were allowed to combine passive elution of antibiotic in a physiological microenvironment with an active release in the presence of bacteria, while the background leaching was not preferred in some cases to avoid drug resistance. Moreover, all these methods have their own limitations and have proven difficult to implement in clinical trials thus far. For example, in the case of pH-responsive elements, many factors can result in changes in local pH values in the body. Further studies will be needed to assess the duration of such a release and the sensitivity of these systems in vivo.

### 3.2. Virulence-Factor-Responsive Systems

Bacteria generate various pathogenic factors in the process of adhesion, aggregation, diffusion and pathogenicity, including various enzymes and toxins, which can also be utilized to design a smart antibacterial DDS. For example, enzyme-responsive polymers, which consist of an enzyme-sensitive group such as an oligopeptide, dipeptide, or tripeptide, undergo changes when triggered by the catalytic action of enzymes, resulting in drug release. Various enzymes, such as HAS and protease, of which their concentration largely increases within the infected microenvironment, have all been explored for the controlled release of antibiotics.

#### 3.2.1. Protease-Triggered Systems

Proteases are the general name of a class of enzymes that hydrolyze protein peptide bonds. They exist widely, mainly in human and animal digestive tracts, and can also be produced by microorganisms. Microbial proteases are mainly produced by mold and bacteria, followed by yeast and actinomycetes.

Based on the strict selectivity of protease to the substrate, investigators have designed a series of protease-responsive DDSs [58,59,60,61,62]. For instance, Johnson et al. engineered lysostaphin encapsulation within protease-degradable hydrogels and subsequent application to infected femurs, which led to fracture callus formation and healing [63]. The inclusion of protease-degradable peptide cross-links in lysostaphin-loaded hydrogels made it possible to deliver lysostaphin on demand in response to infections.

However, the protease-triggered release of drugs was not confined only to the presence of bacteria. The cleavage by the host proteases triggered the undesired release of antibiotics. To make the DDSs more targeted and reduce the accidental release of antibiotics, a many researchers have focused on the response to products of *S. aureus* infections, because of the specificity of their virulence factors [92]. Zhang et al. engineered a titanium coating grafted with vancomycin via a tailor-made peptide that can be cleaved by a *S. aureus*-secreted protease called serine protease-like protease (SplB), allowing the release of vancomycin specifically in the presence of *S. aureus* [62]. The bio-hydrolysis of this peptide was both sensitive and irreversible, highlighting its utility for generating a specific response to *S. aureus* infections.

#### 3.2.2. Hyaluronidase (HAS)-Triggered Systems

Hyaluronidases (HAS) are enzymes that are capable of degrading hyaluronic acid (HA) and hyaluronate. HA constitutes an essential part of the extracellular matrix. Bacteria such as *S. aureus* and *E. coli* utilize HAS as an invasion factor to adhere to the surface of the implants [5]. A previous study has reported that HA-coated mesoporous silica nanoparticles could be degraded upon the addition of HAS [93], making HAS an available trigger for on-demand drug release. Moreover, the secretion of HAS by *S. aureus* and *E. coli* has been studied by Wang et al., where they reported that the increase in local HAS triggered the release of gentamicin from their multilayer films [65]. Similarly, Li et al. designed an intelligent vancomycin-HA-chitosan/β-glycerophophate hydrogel responsive to HAS secretion by *S. aureus* and *S. epidermidis* [64]. The hydrogel possessed antimicrobial properties both in vitro and in vivo that could be modulated by the concentration of HAS.

Interestingly, HA and hyaluronate themselves, which are generally main components of HAS-triggered systems, have been proved capable of decreasing *S. aureus* adhesion and biofilm formation [94], endowing those materials with greater antibacterial potential.

#### 3.2.3. Lipase-Triggered Systems

Lipases are enzymes involved in the digestion of fats to fatty acids and glycerol or other alcohols. They are widely found in animals, plants and microorganisms. With regard to microorganisms, so many pathogenic bacterial species produce lipases that they have been classified as important virulence factors that exert harmful effects in combination with other bacterial enzymes, in particular the phospholipases C [95]. The capacity of lipases to break down ester bonds makes it possible for them to be used as triggers for intelligent drug delivery [96].

PCL microspheres that contain selenium nanoparticles (SeNPs) were developed as a DDS responsive to the existence of *P. aeruginosa* and lipases [68]. It was noticed that the higher lipase titer in vitro led to greater zones of bacterial inhibition. However, researchers compared the release rate of SeNPs in the lipase solution to that in the *P. aeruginosa* cell-free extract, finding that the drug concentration in the medium with bacterial extract was much more prominent. It might indicate more hidden mechanisms when the reactions occurred in vivo. In addition, Shi et al. found that metronidazole linked to dopamine-functionalized PCL nanofiber mats via ester linkage could be triggered to release in response to cholesterol esterase [67]. Furthermore, the release rate of metronidazole increased as the concentration of cholesterol esterase increased. The effective antibacterial capacity of the system indicated that it was a promising bacteria-responsive drug releasing material.

Moreover, one reason as to why bacteria are able to evade the immune system, and thus protect themselves from antibiotics, is that they can survive after phagocytosis by phagocytic cells, especially macrophages, which leads to further infection recurrence [97]. The drug delivery into macrophages is a necessary strategy in improving antibiotic therapy against intracellular infections. Lipase-activated on-demand delivery nanocarriers have been proven to be able to kill intracellular bacteria [72]. Xiong et al. reported a strategy for targeted antibiotic delivery into macrophages via mannose receptors [98,99], utilizing a mannosylated nanogel as the vancomycin carrier responsive to bacterial phospholipase (Figure 4) [71]. The nanogel contained mannosyl ligands conjugated to the shell of the poly(ethylene glycol) arm and polyphosphoester core-crosslinked nanogel. Phosphatase produced by bacteria could degrade the shell, resulting in the vancomycin release. The results suggested that mannosylated nanogels could enter macrophages via the interaction of mannosyl ligands with mannose receptors and release drugs to kill the intracellular bacteria. Similarly, Yang et al. designed a mesoporous silica nanoparticle (MSN) loaded with gentamicin that targeted both planktonic and intracellular infection [69].

#### 3.2.4. Gelatinase-Triggered Systems

Gelatinases, also known as type IV collagenase, belong to the group of metalloproteinases (MMPs) and are able to cause hydrolysis of type IV collagen, leading to the breakdown of the extracellular matrix. A broad spectrum of bacterial species, including Staphylococcus, Enterococcus, and others, is known to produce gelatinases as virulence factors [100]. Based on the activity of gelatinases secreted by bacteria, a gelatin hydrolysis test has been used to distinguish the species of Bacillus, Clostridium, Proteus, Pseudomonas etc. The results revealed that pathogenic bacteria, such as *S. aureus*, were mostly gelatinase-positive [101], which indicated that gelatinase-responsive release of antibiotic agents at the infected site was achievable.

A previous study has proven that gelatinases secreted by *S. aureus* were qualified to activate the release of drugs [102]. Qi et al. further designed an “on-site transformation” system against bacterial infection composed of a chitosan backbone, a PEG-tethered gelatinase-cleavable peptide and an antibacterial peptide KLAK [73]. Cleaved by the gelatinases at the infected sites, the protecting PEG coating disappeared and the conformation changed, subsequently resulting in the release of KLAK peptide. KLAK made contact with the bacterial membranes and killed the bacteria as designed. Similarly, Li et al. [74] developed a kind of core-shell supramolecular gelatin nanoparticle that was capable of delivering vancomycin triggered by gelatinase.

While the smart DDSs are designed to be activated by bacterial enzymes, most antibiotics or AMPs are tied to the skeleton via covalent bond. With the cleavage of specific bonds by enzymes, there is a possibility that some residues are left over on antibacterial agents. The possible residues that remain on the antibiotics may raise the problem of impaired drug activity. It is worth noting that researchers have already pointed out the impact of remaining residues [58]. They demonstrated that nanogels are able to deliver on-demand ciprofloxacin triggered by trypsin, while the groups of the linker residue that remained on the ciprofloxacin negatively affected its efficacy. The antibacterial effect of the nanogels was not as good as that of ciprofloxacin alone. How to solve this impairment is an urgent problem.

Greater consideration should be given to design systems that can efficiently sense and respond to the bacteria and release antibacterial agents in response to enzymatic activities that are unique to the pathogen, while remaining stable against nonspecific cleavages by host enzymes.

### 3.3. Dual Responsive Systems

The concept of the smart bacteria-responsive delivery systems is ideal. As mentioned before, neither the physical changes that occur in the microenvironment nor the production of various enzymes are typically specific. The nonspecific disrupts in these systems present barriers to the success of this approach. To improve the specificity of stimuli-responsive release, researchers focus on the strategy of blending these ideas together. Such dual responsive systems have been reported previously to deliver anti-tumor drugs. Dual-responsive (pH and thermo responsive) nanoparticles from poly(NIPAAm-co-acrylic acid)-b-PCL diblock copolymers were designed to deliver paclitaxel [103]. Drug release was observed only at temperatures greater than 37 °C and at pH conditions between 4 and 6.

Likewise, dual responsive systems can also be used to control the release rate of antibacterial agents during infections. Programmable responsive antibiotics release systems have been investigated, including pH/thermal response and pH/enzymes response models. Chen et al. demonstrated that micelles are sensitive to both pH decreases and lipases [77]. The breakage of the acid-labile linkages led to the release of D-tyrosine to disintegrate the biofilm matrix, while the lipase-triggered breakdown of succinic acid linkages resulted in the release of azithromycin, killing bacteria and destructing biofilms. In addition, Wang et al. designed a vertically aligned mesoporous silica coating on the surface of stainless steel for pH and bacterial lipase-triggered antibiotics release (Figure 5) [76]. It was demonstrated that the lowering of pH triggered the opening of the cyclodextrin valve, enabling the release of the smaller cinnamaldehyde, the first step in killing bacteria. Meanwhile, lipases were shown to cause the cleavage of functionalized cyclodextrin, leading to the release of both cinnamaldehyde and AMPs. This dual release system was shown to inhibit the growth of *S. aureus*, *E. coli*, and MRSA in vitro.

In addition to the systems that combine active releasing modes together, the trigger modes that combine active release and passive release also deserve greater attention. The combination of active “smart” release and passive controlled release can ensure the high concentration of local antibiotics when necessary. The chitosan-graft-polyaniline (CP)/oxidized dextran (OD) hydrogels were proven to be dual responsive to electrical fields and pH [79]. Researchers used amoxicillin as the model drug, and found its release rate increased when an increase in voltage was applied or when the pH decreased. In this way, when it was clear that bacterial infection had occurred, in theory, the rapid release of antibiotics could be realized through the change in artificially applied voltage. The hydrogels presented excellent antibacterial properties and good biocompatibility both in vitro and in vivo, indicating that they are ideal candidates as smart drug delivery vehicles.

### 3.4. Host-Immune-Response-Responsive Systems

Apart from direct changes caused by the bacteria, hosts’ immune responses to infections are also qualified to control the drug release of DDSs. When the infections occur, a large number of immunocytes accumulate nearby and secrete inflammatory factors [104]. The locally elevated levels of such inflammatory factors have aroused great concerns in researchers. Taking matrix metalloproteinases (MMPs) as an example, MMPs comprise a group of endogenous enzymes that play an important role both in physiological and pathological processes, acting on the remodeling and degradation of the extracellular matrix [105]. MMPs have been proven to be associated with the severity of periodontal destruction [106]. Guo et al. designed a degradable MMP8-responsive hydrogel that contained minocycline hydrochloride or AMPs to realize on-demand antibiotics delivery [82]. The results showed that the hydrogel had potential to be used for in situ adaptive degradation in response to peri-implantitis. Similar to MMP8, the reactive oxygen species (ROS) around infected sites also increased significantly. Stavrakis et al. reported a biodegradable coating using a branched poly(ethylene glycol)-poly(propylene sulfide) polymer [60]. The researchers noted a rapid release of antibiotics when using an oxidative environment, confirming a smart active releasing mechanism.

Bone infections will also lead to a series of host responses, including bone resorption. The concentration of acid phosphatase (APS), which takes adenosine triphosphate (ATP) as a substrate, increases significantly in bone infections due to the activation of bone resorption. Polo et al. manufactured a mesoporous bioglass that contained levofloxacin, taking ATP as the molecular gate [83]. Released levofloxacin could only be detected in the presence of APS, ensuring that on-demand release was achieved only due to the specific stimulus typical of a bone infection environment.

The most significant problem of such systems is the lack of specificity to bacterial infections. There are a variety of reasons that can lead to an inflammatory response or bone resorption in vivo, not just bacterial infections. Encapsulating antibiotics into such DDSs may result in the unintended release of antibiotics, which is contrary to what the smart DDSs are designed to do.

## 4. Discussion

As a promising next-generation DDS, smart bacteria-responsive DDSs, which are constructed according to the concept of self-diagnosis to self-treatment, are able to reduce the risk of antibiotic resistance for conventional passive release-based DDSs and remedy the limited delivery range of covalently drug-binding DDSs. In addition, by on-demand release of the smart DDSs, they are able to tackle the problems of external stimuli-responsive release-based antibacterial systems, namely, difficulty in monitoring bacterial growth status around the implants and precisely controlling the appropriate time for external stimuli application. In short, the temporally and spatially on-demand release of free antibacterial agents could help combat infections within a broader peri-implant tissue microenvironment, while mitigating the cytotoxicity associated with the burst release of high doses of physically entrapped antibiotics or risks for developing bacteria resistance due to inadequate or delayed antibiotic releases.

However, the reduction in drug efficiency is a noteworthy problem for both conventional and smart DDSs. During the process of design and manufacture, there are many factors that may impair the drug efficacy. With regard to PEMs, which are commonly used in both conventional and smart DDSs, more than one article reported that the manufacturing methods of PEMs affected the rate of responsive release. Zhuk et al. reported that PEMs manufactured by the spin-assisted LBL technique showed both a slower and lower release compared to dip-deposited films [52]. However, the research by Zhou et al. suggested otherwise, indicating that the antibacterial activities of dip-deposited films were limited [30]. Moreover, current experimental results have indicated that the retention/release properties of PEMs are highly dependent on the selection and matching degree of polyelectrolytes and antimicrobials. Greater caution is required when choosing manufacturing methods and corresponding antibiotics.

Although in theory, the design concept of smart bacteria-responsive DDSs is to effectively treat peri-implant infections, there are still various problems that exist in practice. DDSs triggered by pH and temperature switches, in most cases, combined passive elution and active release, while the background leaching was not preferred in some cases to avoid drug resistance. In addition, there were multiple causes for the change in physical microenvironment, which indicated that the non-specific trigger would be a major issue. This also happened in host-immune-response-responsive systems. For virulence-factor-responsive DDSs, the specificity was improved to some extent, but enzymes such as protease and lipase could be derived from the host. To conclude, although the above DDSs can theoretically be stimulated as designed, the trigger of these systems are not specific to infection. The lack of specificity is a common problem in the smart DDSs to date. Due to the complexity in vivo, researchers cannot guarantee DDS’ stability under the accidental non-specific stimuli, which is undesirable. In the case of double or multiple-responsive systems, the situation becomes even more uncontrollable.

It is believed that every kind of system has its pros and cons. Inspired by smart DDSs designed particularly for MRSA [75,107] and *S. aureus* [92], finding specific substrates as triggers for various infections is considered to be promising. Just as specific virulence factors exist in MRSA-infected microenvironments, if other infected microenvironment-specific factors can be found, the problem regarding specificity can be improved and truly smart DDSs can be achieved.

In addition to improving the specificity of DDSs, it is believed that giving DDSs more functionality is one of the promising directions. First of all, smart DDSs can be designed to target intracellular infections. As is well known, intracellular bacteria are among the most dangerous causes of drug resistance. Bacteria engulfed by macrophages are able to escape from antibiotic attacks, because challenges remain in intracellular drug delivery specific to bacteria-infected cells and efficient uptake into intracellular bacteria. In this case, smart DDSs are clearly one of the best methods to deliver antibiotics directly to infected cells, such as macrophages, which may be achieved by adding specific antibodies to the surface of the system to target specific cells and utilizing materials such as nanoparticles that can be swallowed by cells to kill the intracellular bacteria. Several studies have proven its feasibility [71,72].

In addition, there are often other demands, such as anti-inflammation and bone regeneration, that must be realized when DDSs are implanted. Some of the DDSs were able to implement multiple functions at the same time. In fact, smart DDSs are also commonly used in other therapies, such as anti-inflammatory or anti-tumor therapies. Therefore, multifunctional smart DDSs are feasible, in which antibacterial properties can be achieved by the component of DDSs, such as metal nanoparticles, and the loaded drugs solve other problems. These ideas can be widely extended. For example, it is more desirable if the smart DDSs composed of hydrogels designed for oral implantation promote tissue regeneration and TNTs facilitate osseous integration. Moreover, bacterial infections are often accompanied by subsequent inflammatory responses, so smart anti-bacteria and anti-inflammation DDSs are preferred for sequential treatment [81]. The pattern of on-demand drug release can effectively circumvent drug resistance of all drugs loaded on DDSs, making it suitable for versatile drug delivery.

Overall, smart bacteria-responsive DDSs are believed to be the next generation of mainstream antibacterial therapy. The need for multimodality strategies that take into account various stages of pathogenesis [108] to improve specificity, while minimizing the negative impact on peri-implant tissues or encouraging implant-tissue integrations, is increasingly recognized. Targeted smart bacteria-responsive DDSs or versatile DDSs are believed to be promising. More attention should be paid to this area.

## Figures and Tables

**Figure 1 jfb-13-00173-f001:**
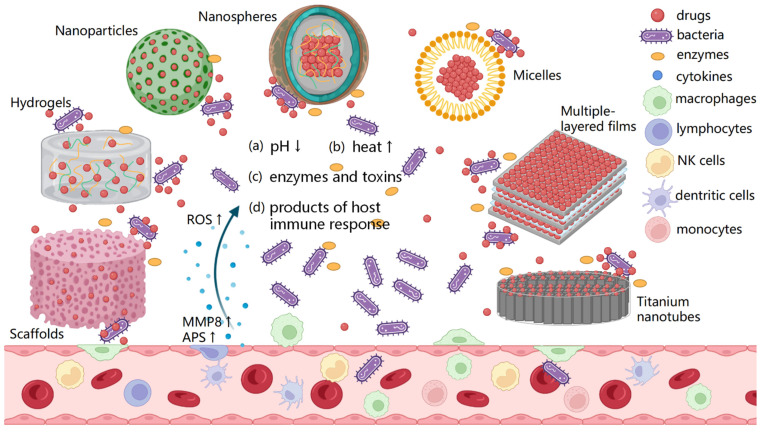
The schematic representation of smart bacteria-responsive drug delivery systems. Scaffolds, hydrogels, nanoparticles, nanosphere, micelles, multiple-layer films and titanium nanotubes (TNTs) loaded with drugs are triggered by the changes specific to the infection microenvironment, including the (a) pH decreasing, (b) elevated local temperature, (c) bacteria-specific enzymes and toxins and (d) products of host immune response, aiming to kill the bacteria.

**Figure 2 jfb-13-00173-f002:**
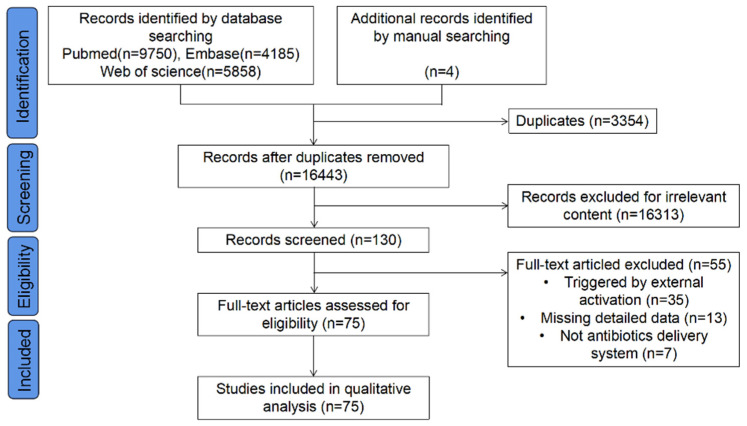
Search flowchart.

**Figure 3 jfb-13-00173-f003:**
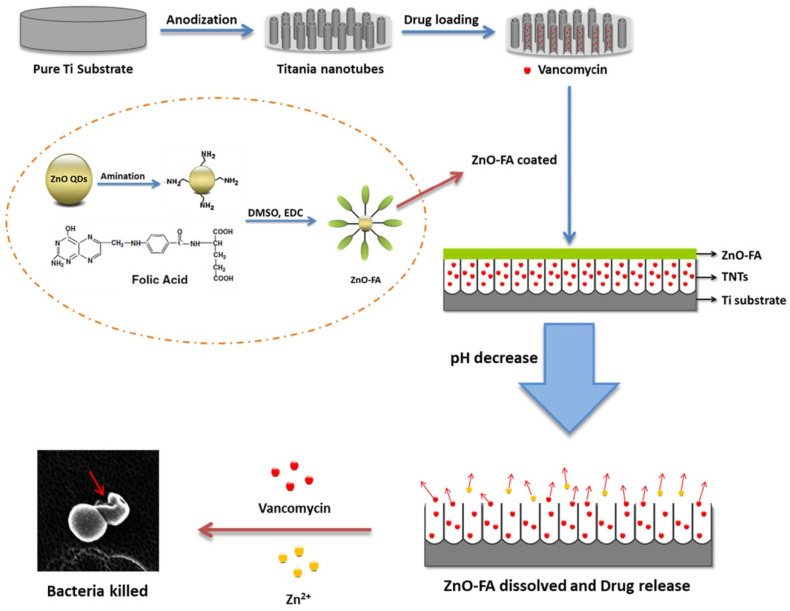
The schematic illustration of fabrication process of TNTs-Van@ZnO-FA system and synergistic bacteria-killing triggered by pH. Reproduced with the permission from ref. [23]. Copyright 2015 Elsevier.

**Figure 4 jfb-13-00173-f004:**
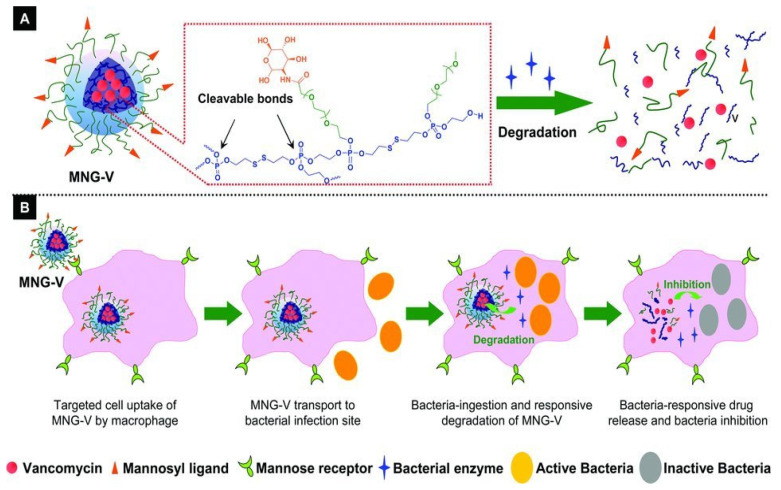
(**A**) Schematic illustration of a vancomycin-loaded mannosylated nanogels (MNG-V) and the bacteria-responsive drug release. (**B**) Schematic illustration of targeted uptake of MNG-V, transport, degradation, drug release and bacteria inhibition. Reproduced with the permission from ref. [52]. Copyright 2012 John Wiley and Sons.

**Figure 5 jfb-13-00173-f005:**
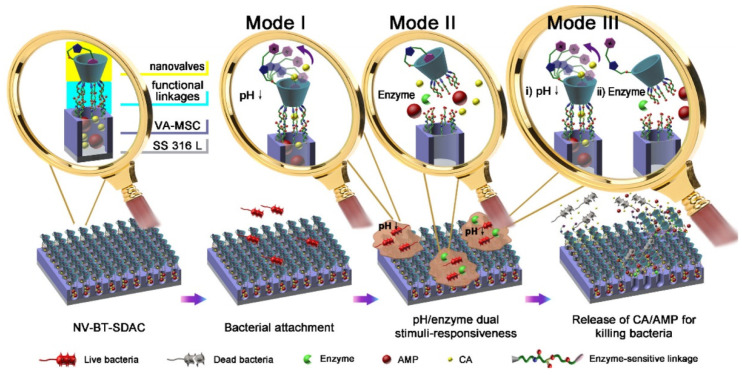
Schematic representation of structure and working mechanisms for NV-BT-SDACz deposited on SS316L. Reproduced with the permission from ref. [55]. Copyright 2017 American Chemical Society.

**Table 1 jfb-13-00173-t001:** Features of the included studies.

Study	Drug(s)	Trigger(s)	Type	Structures	Bacteria	Outcome(s)
Li D. et al., 2022 [10]	VAN	pH switches	Nanoparticles	VAN@PEG-VAN	*S. aureus*	ZOI, MCB (in vitro), BIR (in vivo)
Fu M. et al., 2022 [11]	GTM	pH switches	Hydrogels	GTM@P(AA-co-HEMA)	*E. coli*, *S. aureus*	DLC, ZOI, BIR (in vivo)
Yang, L. et al., 2022 [12]	TOB	pH switches	Films	TOB@Protocatechualdehyde-aminoglycosides	*E. coli*, *P. aeruginosa*, *S. epidermidis*, *S. aureus*	ZOI, MCB (in vitro), BIR (in vivo)
Cámara-Torres M. et al., 2021 [13]	CFX, GTM	pH switches, ion exchange	Scaffolds	PEOT/PBT-MgAl-CFX, PEOT/PBT-Zrp-GTM	*S. epidermidis*, *P. aeruginosa*	ZOI
Guo, R. et al., 2021 [14]	Triclosan	pH switches	Micelles	PLA-PEG-PAE	*E. coli*, *S. aureus*	BIR (in vitro), MCB (in vitro)
Ramesh, S. et al., 2021 [15]	ZnONPs	pH switches	Hydrogels	ZnONPs@CS-GP	*E. coli*, *S. aureus*	BIR (in vitro), DLC
Sang, S. et al., 2021 [16]	GTM	pH switches	Films	GTM-Silk protein	*E. coli*, *S. aureus*	ZOI, DLC, BIR (in vitro), MCB (in vitro)
Yan K. et al., 2021 [17]	AgNPs	pH switches	Hydrogels	CS-AgNPs	*E. coli*, *S. aureus*	ZOI, MCB (in vitro)
Zha, J. et al., 2021 [18]	Curcumin	pH switches	Hydrogels	Curcumin@POEGMA-PEI	MRSA	BIR (in vitro)
Hassan D. et al., 2020 [19]	VAN	pH switches	Quatsomes	VAN-StBAclm	MRSA	DLC, BIR (in vitro and in vivo), MCB (in vitro)
Li, M. et al., 2020 [20]	GTM	pH switches	Films	GTM@(al-ALG/PEI)_10_	*E. coli*, *S. aureus*	ZOI, DLC, BIR (in vitro), MCB (in vitro)
Chen J. et al., 2020 [21]	AMPs	pH switches	Films	TNTs-PMAA-AMP	*S. aureus*, *E. coli*, *P. aeruginosa*, MRSA	BIR (in vitro and in vivo)
Tao B. et al., 2019 [22]	GTM	pH switches	Films	TNTs-BMP2-(ADA-GTM/CS)_10_	*S. aureus*, *E. coli*	BIR (in vitro), MCB (in vitro)
Jin, X. et al., 2019 [23]	GTM	pH switches	Scaffolds	Porous hydroxyapatite-GTM	*E. coli*, *S. aureus*	DLC, ZOI, BIR (in vitro and in vivo)
de Avila E.D. et al., 2019 [24]	TC	pH switches	Films	(PAA/PLL-TC)_10_	*P. gingivalis*	BIR (in vitro)
Cao J. et al., 2019 [25]	CHX	pH switches	Nanoparticles	CHX@PMPC-CS	*S. aureus*	BIR (in vitro); MCB (in vitro)
Hu J. et al., 2019 [26]	TOB, ornidazole	pH switches	Hydrogels	TOB-G1-orni	*S. aureus*, *P. aeruginosa*, *C. sporogenes*, *B. fragilis*	BIR (in vitro); MCB (in vitro)
Karakeçili A. et al., 2019 [27]	VAN	pH switches	Nanoparticles	ZIF8/VAN	*S. aureus*	DLC, BIR (in vitro)
Maji R. et al., 2019 [28]	VAN	pH switches	Nanoparticles	VAN@lipid–dendrimer hybrid NPs	MRSA	DLC, BIR (in vitro)
Mir M. et al., 2019 [29]	CAR	pH switches	Nanoparticles	CAR@PCL-NPs	MRSA	BIR (in vitro)
Zhou W. et al., 2018 [30]	GTM, AgNPs	pH switches	Films	CS-(AgNPs/GTM-SF)	*S. aureus*	BIR (in vitro)
Xiang Y. et al., 2018 [31]	VAN	pH switches	Quantum dots	TNTs-VAN@ZnO-FA	*S. aureus*	BIR (in vitro); MCB (in vitro)
Placente D. et al., 2018 [32]	CFX	pH switches	Nanoparticles	Lipid membrane mimetic coated nano-hydroxyapatite	*E. coli*, *P. aeruginosa*, *S. aureus*	BIR (in vitro), ZOI
Cicuéndez M. et al., 2018 [33]	LFX	pH switches	Scaffolds	MGHA-LFX	*S. aureus*	BIR (in vitro), MCB (in vitro)
Dai T. et al., 2018 [34]	AgNPs	pH switches	Hydrogels	Dex-G5-AgNPs	*E. coli*, *P. aeruginosa*, *S. aureus*, *S. epidermidis*	BIR (in vitro and in vivo), MCB (in vitro)
Dubovoy V. et al., 2018 [35]	BAC	pH swtiches	Nanoparticles	BAC-MSNs	*S. aureus*	BIR (in vitro)
Mhule D. et al., 2018 [36]	VAN	pH swtiches	Nanoparticles	VAN@NMEO	MRSA	DLC, BIR (in vitro and in vivo)
Soltani B. et al., 2018 [37]	GTM	pH swtiches	Nanoparticles	GTM@nanoscale zeolitic imidazolate frame-work-8	*E. coli*, *S. aureus*	DLC, BIR (in vitro)
Yu X. et al., 2018 [38]	VAN	pH swtiches	Microspheres	PLGA–NaHCO3–Van	*S. aureus*, *MRSA*	DLC, BIR (in vitro), ZOI
Zhang S. et al., 2018 [39]	AgNPs	pH switches	Films	AgNPs@PLL/PG	*S. aureus*	BIR (in vitro)
Zhou, W. et al., 2018 [40]	TOB	pH switches	Films	TOB@ (CHT/HET)_2_	*S. aureus*	ZOI, BIR (in vitro), MCB (in vitro)
Wang T. et al., 2017 [41]	VAN, Ag	pH switches	Films	TNT(NH2)-VAN@Zn-BIX, TNT(NH2)-Ag@Zn-BIX	*E. coli*, *S. aureus*	ZOI, BIR (in vitro), MCB (in vitro)
Pamfil D. et al., 2017 [42]	CFX	pH switches	Hydrogels	CFX @HEMA/C-CA	*S. aureus*	ZOI
Liu Z. et al., 2017 [43]	VAN	pH switches	Nanoparticles	VAN@PVA/PLGA	*S. aureus*	ZOI, MCB (in vitro)
Dong Y. et al., 2017 [44]	AgNPs	pH switches	Films	TNTs-acetal linker-AgNPs	*E. coli*, *S. aureus*	BIR (in vitro)
Kalhapure R. S. et al., 2017 [45]	VAN	pH switches	Nanoparticles	VAN@(2-(2,4,6trimethoxyphenyl)-1,3-dioxane-5,5-diyl) bis(methylene) distearate	*S. aureus*, MRSA	DLC, BIR (in vitro and in vivo)
Sang Q et al., 2017 [46]	CFX	pH switches	Nanofibers	Gelatin-sodium bicarbonate	*E. coli*, *S. aureus*	DLC, BIR (in vitro)
Onat B. et al., 2016 [47]	Triclosan	pH switches	Micelles	(Triclosan@βPDMA-b-PDPA)3	*S. aureus*, *E. coli*	BIR (in vitro), ZOI
Fullriede H. et al., 2016 [48]	CHX	pH switches	Nanoparticles	CHX@silica nanoparticles-PVP	*S. aureus*, *S. mutans*	BIR (in vitro)
Kalhapure R. S. et al., 2017 [49]	VAN	pH switches	Nanoparticles	CS@VAN-AGS	MRSA	BIR (in vitro and in vivo)
Anandhakumar S. et al., 2016 [50]	CFX	pH switches	Films	(PAH/PMAA-CFX)8	*E. coli*	ZOI
Zhang Z. et al., 2015 [51]	MNC	pH switches	Films	(DS-Mg2+-MNC)-GA	*E. coli*, *S. aureus*	BIR (in vitro)
Zhuk I. et al., 2014 [52]	GTM, TOB, polymyxin B	pH switches	Films	TA-GTM/TOB/polymyxin B (PolyB)	*S. epidermidis*, *S. aureus*	BIR (in vitro), MCB (in vitro)
Zhang Z. et al., 2014 [53]	MNC	pH switches	Films	(DS-Ca2+/MNC-Ca2+/GA-Ca2+)8	*E. coli*, *S. aureus*, MRSA, *S. epidermidis*	BIR (in vitro), MCB (in vitro)
Pichavant, L. et al., 2012 [54]	GTM	pH switches	Nanoparticles	GTM@Functionalized PEO	*S. aureus*	DLC, BIR (in vitro)
Choi H. et al., 2021 [55]	LFX	High temperatures	Films	Ti-PDEGMA-LFX	*S. aureus*	DLC, MCB (in vitro and in vivo)
Li B. et al., 2021 [56]	Glycerin, simvastatin	High temperatures	Hydrogels	TNTs-CS-glycerin-hydroxypropylmethyl	*E. coli*, *S. aureus*	BIR (in vivo)
Liang J. et al., 2019 [57]	Colistin, AMPs	Bacterial contact	Microgels	PAA-colistin, PAA-AMPs	*E. coli*, *S. epidermidis*	BIR (in vitro)
Bourgat Y. et al., 2021 [58]	CFX	Enzymes (PS)	Nanogels	Alginate-PLL-CFX	*S. aureus*	BIR (in vitro)
Timin A. et al., 2018 [59]	CFS	Enzymes (PS)	Scaffolds	PCL-CFS, PHB-CFS, (PHB-PANi)-CFS	*E. coli*	ZOI
Liao X. et al., 2021 [60]	CHX	Enzymes (PS)	Films	(MTT/PLL-CHX)_10_	*S. aureus*	ZOI, DLC, BIR (in vitro and in vivo),
Yu X. et al., 2021 [61]	VAN	Enzymes (PS)	Films	(MTT/PLL-VAN)_8_	*S. aureus*	ZOI, MCB (in vitro), BIR (in vitro and in vivo)
Zhang Y. et al., 2021 [62]	VAN	Enzymes (PS)	Films	Ti-SRP1 peptides-VAN	*S. aureus*	BIR (in vitro)
Johnson CT. et al., 2018 [63]	Lysostaphin	Enzymes (PS)	Hydrogels	PEG-4MAL-lysostaphin	*S. aureus*	BIR(in vitro and in vivo)
Li Y. et al., 2020 [64]	VAN	Enzymes (HAS)	Hydrogels	VAN-HA-CS/β-glycerophosphate	*S. aureus*, *S. epidermidis*	BIR (in vitro)
Wang B. et al., 2018 [65]	GTM	Enzymes (HAS)	Films	(MMT/HA-GTM)_10_	*S. aureus*, *E. coli*	BIR (in vitro and in vivo); ZOI; MCB (in vitro and in vivo)
Ji H. et al., 2016 [66]	AA, MNPs	Enzymes (HAS)	Nanosheet	AA@GMSN-HA-MNPs	*S. aureus*, *E. coli*	BIR (in vitro and in vivo), MCB (in vitro)
Shi R. et al., 2019 [67]	MNA	Enzymes (LS)	Films	PCL-dopamine-MNA	*H. pylori*	BIR (in vitro)
Filipović N. et al., 2019 [68]	SeNPs	Enzymes (LS)	Microspheres	PCL-SeNPs	*S. epidermidis*, *S. aureus*	ZOI, DLC
Yang S. et al., 2018 [69]	GTM	Enzymes (LS)	Nanoparticles	GTM@MSNs-lipid-UBI	*S. aureus*	BIR (in vitro and in vivo)
Li Y.M. et al., 2017 [70]	Triclosan, AMP, parasin I, lysozyme	Enzymes (LS)	Micelles	PEG-b-PA/PN@drugs	*S. aureus*, *E. coli*	DLC, BIR (in vitro)
Xiong M. et al., 2012 [71]	VAN	Enzymes (LS)	Nanogels	Mannosyl-PEG-polyphosphoester-VAN	*S. aureus*	DLC, BIR (in vitro and in vivo), MCB (in vivo)
Xiong M. et al., 2012 [72]	VAN	Enzymes (LS)	Nanogels	PEG-PCL-polyphosphoester-VAN	*S. aureus*	DLC, BIR (in vitro)
Qi GB. et al. 2017 [73]	AMPs	Enzymes (GS)	Nanoparticles	CS-CPC1-AMPs	*S. aureus*	BIR (in vitro and in vivo), MCB (in vitro and in vivo)
Li L.L. et al., 2014 [74]	VAN	Enzymes (GS)	Nanoparticles	SGNPs-VAN @RBC	*S. epidermidis*, *S. aureus*	DLC, BIR (in vitro), MCB
Tonkin, R. L., et al., 2022 [75]	VAN	Cytoloytic toxin	Capsosomes	VAN@Mesosilica-PAH-(PMAA-PDA/liposome)_3_	MRSA	ZOI, survival rate
Wang T. et al., 2017 [76]	Ampicillin, CA	pH switches, enzymes (LS)	Films	VAMSC-CA/ampicillin-monopyridine functionalized β-cyclodextrin	*E. coli*, *S. aureus*, MRSA	BIR (in vitro), MCB (in vitro)
Chen M. et al., 2019 [77]	D-tyrosine, AZM	pH switches, enzymes (LS)	Micelles	DOEAz-tyrosine	*P. aeruginosa*	BIR (in vitro and in vivo), MCB (in vitro and in vivo)
Chen M. et al., 2018 [78]	VAN, CFX	pH switches, enzymes (LS)	Micelles	CFX@VAN-PECL	*P. aeruginosa*	BIR (in vitro and in vivo), MCB (in vitro and in vivo)
Qu J. et al., 2018 [79]	AMX	pH and electric field switches	Hydrogels	CP/OD-AMX	*E. coli*, *S. aureus*	BIR (in vitro)
Stanton M. M. et al., 2017 [80]	CFX	pH and external magnetic guidance	Biohybrids	MSR1-CFX@MSM	*E. coli*	BIR (in vitro), MCB (in vitro)
Hu C. et al., 2020 [81]	AMIK, naproxen	pH switches, ROS	Hydrogels	(AMIK@ALG-BA)-(naproxen@HA-cholesterol)	*S. aureus*, *P. aeruginosa*	DLC, BIR (in vitro and in vivo), MCB (in vitro)
Guo J. et al., 2019 [82]	MNC, AMP	MMP-8	Hydrogels	MNC@(4-arm PEG-diacrylate)-MMP8 sensitive peptide	*P. gingivalis*	BIR (in vitro)
Polo L. et al., 2018 [83]	LFX	APS	Scaffolds	MBG-LFX-ATP	*E. coli*	BIR (in vitro)
Stavrakis A. et al., 2016 [84]	VAN, TGC	ROS	Films	Van@PEG-PPS, TGC@PEG-PPS	*S. aureus*	BIR (in vivo)

TOB: tobromycin; CFX: ciprofloxacin; GTM: gentamicin; AgNPs: silver nanoparticles; BAC: benzalkonium chloride; AMPs: antimicrobial peptides; TC: tetracycline; VAN: vancomycin; LFX: levofloxiacin; CHX: chlorhexidine; MNC: minocycline; CFS: ceftriaxone sodium; AA: ascorbic acid; MNPs: ferromagnetic nanoparticles; MNA: metronidazole; SeNPs: selenium nanoparticles; CA: cinnamaldehyde; AZM: azithromycin; AMX: amoxicillin; TGC: tigecycline; CAR: carvacrol; AMIK: amikacin; PS: protease; HAS: hyaluronidase; LS: lipase; GS: gelatinase; APS: acid phosphatase; al-ALG: aldylated sodium alginate; ROS: reactive oxygen species; PEOT: poly(ethyleneoxideterephthalate); PBT: poly(butyleneterephthalate); MgAl: magnesium aluminum layered double hydroxides; ZrP: α-zirconium phosphates; PLA: poly(lactic-co-glycolic acid); PLE: poly(β-amino ester); PEG: poly(ethylene glycol); PAE: poly(β-amino ester); GP: glycerol phosphate; CS: chitosan; POEGMA: poly(oligo(ethylene glycol) methacrylate); PEI: poly(ethyleneimine); TNTs: TiO2 nanotubes; PMPC: poly(2methacryloyloxyethyl phosphorylcholine); SGNPs: supramolecular gelatin nanoparticles; G1-orni: amine-terminated poly(amidoamine); PMAA: poly(methacrylic acid); ADA: alginate dialdehyde; PAA: poly(acrylic acid); PLL: poly-l-lysine; SF: silk fibroin; FA: folic acid; MGHA: nanocrystalline apatite uniformly embedded into a mesostructured SiO2-CaO-P2O5 glass wall; CHT: positively charged chitosan; HET: heparin miscalls; BIX: 1,4-bis(imidazol-1-ylmethyl) benzene; HEMA: 2-hydroxyethyl methacrylate; C-CA: citraconic anhydride-modified collagen; PVA: poly(vinyl alcohol); PLGA: poly(lactide-glycolide acid); βPDMA-b-PDPA: poly[3-dimethyl (methacryloyloxyethyl) ammonium propane sulfonate-b-2-(diisopropylamino)ethyl methacrylate]; PG: poly-L-glutamic; NMEO: N-(2-morpholinoethyl) oleamide; AGS: anionic gemini surfactant; PVP: poly(4-vinylpyridine); PAH: poly(allylamine hydrochloride); DS: dextran sulfate; GA: gelatin type A; TA: tannic acid; PEO: poly(ethylene oxide); PDEGMA: poly(di(ethylene glycol) methyl ether methacrylate); PCL: polycaprolactone; PHB: poly(3-hydroxybutyrate); PANi: polyaniline; MTT: montmorillonite; GMSN: graphene-mesoporous silica nanosheet; PEG-4MAL: four-arm PEG macromers functionalized with terminal maleimide groups; HA: hyaluronic acid; GNT; MMT: montmorillonite; MSNs: mesoporous silica nanoparticle;UBI: ubiquicidin; CP: chitosan-graft-polyaniline; PMA-PDA: poly(methacrylic acid) functionalized with pyridine dithioethylamine; VAMSC: vertically aligned mesoporous silica coating; PPS: poly(propylene sulfide); OD: oxidized dextran; MBG: mesoporous bioglass; PECL: poly(ethylene glycol)−poly(ε-caprolactone); PEG-b-PA: poly(ethylene glycol)-b-poly(2-((((4-acetoxybenzyl)oxy)carbonyl)amino)ethyl methacrylate); PEG-b-PN: poly(ethylene glycol)-bpoly(2-((((4-nitrobenzyl)oxy)carbonyl)amino)ethyl methacrylate); MSR1: magnetosopirrillum gryphiswalense; MSM: mesoporous silica microtube; ALG-BA: alginate-boronic acid; *S. aureus*: Staphylococcus aureus; *S. epidermidis*: Staphylococcus epidermidis; *P. aeruginosa*: Pseudomonas aeruginosa; *E. coli*: Escherichia coli; MRSA: methicillin-resistant Staphylococcus aureus; *H. pylori*: Helicobacter pylori; *P. gingivalis*: prophyromonas gingivalis; *S. mutans*: Staphylococcus mutans; BIR: bacterial inhibition rate; DLC: drug leakage concentration; ZOI: zones of bacterial inhibition; MCB: morphological characterization of bacteria.

## Data Availability

Not applicable.

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
