# Peer review of "Smart Bacteria-Responsive Drug Delivery Systems in Medical Implants"

_jfb, 2022, doi:10.3390/jfb13040173_

Round 1

Reviewer 1 Report

The review is nicely summarizing recent advances in bacteria-responsive drug delivery systems used for combating bacterial infections, mainly according to the different trigger modes including physical stimuli-responsive, virulence-factor-responsive, host-immune-response responsive and their combinations. The work is well organized and clearly written. The study includes relevant references covering the field. The manuscript will be an excellent contribution after some minor revision.

I have the only question after reading the manuscript. The authors have reviewed the stimuli-responsive systems intended to deliver antibiotics. However, as far as I know, the emerging resistance of bacteria to conventional antibiotics appears to be an issue in combating bacterial infections. With this regard, metallic nanoparticles are actively studied considering their antimicrobial properties. This is exampled by silver and gold nanoparticles. I would like the authors to discuss the feasibility of nanoprticles to be included in such systems. Along the antimicrobial effect, the nanoprticles may also bring the response to external stimuly to delivery systems. This may be useful trigger an on-demnad drug release. As I have noticed, authors have not considered "nanoparticles" in their search terms.

Author Response

Thank you for your question.

Response1: Firstly, metallic nanoparticles are indeed a very potential material which can make up the smart anti-bacterial DDSs. They are often added to all kinds of responsive biomaterials because of their antimicrobial properties. In this case, they are most commonly used in anti-cancer systems, in which most of them are not applied as implants, being excluded from our research. Additionally, we have already include several studies regarding metallic nanoparticles in our research before, such as the study of Yan K. et al. 2021, which added silver nanoparticles to the hydrogels. The antibacterial ability was performed via silver nanoparticles.

Response 2:  Secondly, we added ”nanoparticles”to our search terms, and re-screened them. We found more articles proving that metallic nanoparticles are feasible in such systems.

Response 3: Thirdly, we have taking DDSs triggered by the combination of external and autonomous stimuli into account. However, we haven’t found such systems containing nanoparticles. 

Reviewer 2 Report

In this review article, the authors have discussed about the stimuli responsive drug delivery systems for bacterial infections. This review is well organized and discussed. Also, this is insightful and would helpful to the readers working in this research field. So, I would recommend publication of this manuscript in this journal. 

Author Response

Thank you for your comments. We are glad to get your comments.

Reviewer 3 Report

The paper entitled Smart bacteria-responsive drug delivery systems in medical implants sumarizes recent advances in bacteria respnsive drug delivery systems. However the major concern is that authors gathered knowledge in this field but give no future lead courses in this field. Authors should review literature and future perspectives, and give critical guidelines.

Author Response

Thank you for your suggestions. We revised our article and added several future lead courses in the discussion.

Firstly, we still consider improving specificity of substrates as the most promising future in this field. Secondly, we believe that the use of the smart DDSs to target intracellular infections is highly promising because intracellular bacteria are among the most dangerous causes of drug resistance. Last but not least, multifunctional smart DDSs are feasible, in which antibacterial properties can be achieved by the component of DDSs, such as metal nanoparticles, with the loaded drugs solving other problems. Versatile DDSs which can realize the ability of anti-bacteria and anti-inflammation or promoting tissue regeneration are recommended.

Round 2

Reviewer 3 Report

.